# One-Stage Soft Tissue Reconstruction Following Sarcoma Excision: A Personalized Multidisciplinary Approach Called “Orthoplasty”

**DOI:** 10.3390/jpm10040278

**Published:** 2020-12-14

**Authors:** Andrea Angelini, Cesare Tiengo, Regina Sonda, Antonio Berizzi, Franco Bassetto, Pietro Ruggieri

**Affiliations:** 1Department of Orthopedics and Orthopedic Oncology, University of Padova, 35122 Padova, Italy; andrea.angelini@unipd.it (A.A.); antonio.berizzi@unipd.it (A.B.); 2Clinic of Plastic Surgery, University of Padova, 35122 Padova, Italy; cesare.tiengo@unipd.it (C.T.); sondaregina89@gmail.com (R.S.); franco.bassetto@unipd.it (F.B.)

**Keywords:** reconstruction, complications, bone tumor, soft tissue sarcoma, flaps, pelvis, function

## Abstract

**Background and Objectives**. Wide surgical resection is a relevant factor for local control in sarcomas. Plastic surgery is mandatory in demanding reconstructions. We analyzed patients treated by a multidisciplinary team to evaluate indications and surgical approaches, complications and therapeutic/functional outcomes. **Methods**. We analyzed 161 patients (86 males (53%), mean age 56 years) from 2006 to 2017. Patients were treated for their primary tumor (120, 75.5%) or after unplanned excision/recurrence (41, 25.5%). Sites included lower limbs (36.6%), upper limbs (19.2%), head/neck (21.1%), trunk (14.9%) and pelvis (8.1%). Orthoplasty has been considered for flaps (54), skin grafts (42), wide excisions (40) and other procedures (25). **Results**. At a mean follow-up of 5.3 years (range 2–10.5), patients continuously showed no evidence of disease (NED) in 130 cases (80.7%), were alive with disease (AWD) in 10 cases (6.2%) and were dead with disease (DWD) in 21 cases (13.0%). Overall, 62 patients (38.5%) developed a complication (56 minor (90.3%) and 6 major (9.7%)). Flap loss occurred in 5/48 patients (10.4%). The mean Musculoskeletal Tumor Society (MSTS) and Toronto Extremity Salvage Score (TESS) was 74.8 ± 14 and 79.1 ± 13, respectively. **Conclusions**. Orthoplasty is a combined approach effective in management of sarcoma patients, maximizing adequate surgical resection, limb salvaging and functional recovery. One-stage reconstructions are technically feasible and are not associated with increased risk of complications.

## 1. Introduction

Before the advent of chemotherapy, the primary surgical treatment for bone tumors of extremity was amputation. To date, thanks to the advances in adjuvant treatments and surgical techniques, limb salvage surgery has proven feasible with adequate margins in over 90% of patients with primary and secondary bone tumors as well as for soft tissue sarcoma (STS) of the limbs [1,2]. In the presence of tumors with a large soft tissue extension, the limb-preserving approach often creates complex defects that are challenging for wound closure and functional results.

Historically, orthopedic and plastic surgeons have worked separately when faced with challenging reconstructive cases involving skeletal and soft tissue reconstruction of a lower extremity. Pioneers in the field of musculoskeletal oncology have emphasized and anticipated the essential role of a multidisciplinary approach in the management of bone and soft tissue sarcomas. With time, separate skill sets and knowledge have been merged in a positive collaboration (called orthoplasty) to offer patients the best possible surgical approach (chance for success). The involvement of plastic surgeons in preoperative planning and surgery has been increasing over the last decade, thanks to the numerous techniques available in the reconstructive ladder. There are approximately three main goals of plastic surgery in musculoskeletal oncology: (1) refinements in surgical techniques have resulted in the development of function-preserving approaches avoiding limb amputation with the simultaneous increased need of flap reconstruction [3]. (2) A post-surgical stable wound healing with a quicker start of adjuvant radiotherapy is required, forcing the oncologic team to ensure more suitable soft tissue coverage; (3) it is important to reduce the risk of postoperative complications (infection or delayed wound healing) after prosthetic reconstruction for bone tumors as they represent the most frequent cause of implant failure [4,5,6].

We performed a retrospective analysis of patients with bone and soft tissue sarcomas treated with resection and immediate soft tissue reconstruction in a multidisciplinary team (orthopedic and plastic surgeons) in order to evaluate the (1) indications and surgical approach, (2) complications related to flap reconstruction and (3) therapeutic and functional outcomes of these patients at mid- and long-term follow-up.

## 2. Materials and Methods

We analyzed a prospectively collected database of patients with bone and soft tissue tumors treated at our institute from 2006 to 2017. Our hospital is a tertiary referral hospital with a specialized musculoskeletal tumor center and a multidisciplinary team that includes vascular, microvascular and plastic surgeons. We included all patients with oncologic disease treated with an interdisciplinary approach that combines the expertise of the orthopedic oncology team and plastic surgeons. Preoperative consultations with plastic surgeons were arranged whenever a flap reconstruction was considered mandatory to achieve adequate wound closure, to fill surgical dead space or to protect critical structures (i.e., nerves, tendons, vessel and/or bone). Patients with less than 2 years of follow-up, no postoperative functional records and those lost at follow-up were excluded. The research has been performed in accordance with the declaration of Helsinki. All patients or their relatives gave written informed consent to be included in scientific studies at admission to the hospital. As this analysis consists of anonymized clinical routine data, Research Ethics Committee Approval was not necessary in our institution.

Patient age, gender, comorbidities, tumor characteristics (histotype, grade, staging, tumor depth and anatomic site), preoperative history and pre-surgical status, imaging evaluation, surgical data (timing of reconstruction, bone resection, flap details and additional procedures), adjuvant treatments (radiation, chemotherapy), oncologic outcome, complications and their applied treatment were evaluated. Tumor depth was classified as superficial (above superficial fascia without infiltration) or deep. The pre-surgical status included “primarily treated”, “referral after unplanned excision” and “referral due to local recurrence”. For patients who had a previous unplanned excision, re-excision was performed by removing the tumor bed to achieve a wide margin [7]. The surgical re-excision margin was determined based on enhanced magnetic resonance imaging (MRI) performed just before re-excision, which was undertaken with a wide margin of normal surrounding tissues. Regarding imaging evaluation, all patients underwent preoperative plain radiography, computerized tomography (CT) and magnetic resonance imaging (MRI) of the affected bone segment. CT angiograms were used to visualize the vascular supply of the segment and extremity, to define inflow, outflow and interruptions in blood flow. When a pedicled or free perforator flap was required, the vascular network was studied with particular attention in a multidisciplinary meeting. Any concern for vascular injury or preexisting pathology warranted a consultation with a vascular surgeon. The flap reconstructions were divided into local flaps (advancement, rotation and transposition), regional flaps (freed tissue is moved underneath normal tissue) and free microsurgical flaps. Radiation therapy was administered based on histologic diagnosis, surgical margins and in patients with a higher risk of local recurrence.

Routine follow-up examinations were performed with ambulatory checks every 3 months during the first 3 years after surgery, then every 4 months in the fourth year, then every 6 months for 1 year and then once annually. Follow-up evaluations included physical examination and functional evaluation, imaging studies and disease-specific chest imaging. Oncologic results were evaluated according to local recurrence, metastasis or death, and patients were classified as follows: (1) no evidence of disease at the latest routine check (NED); (2) disease free after treatment of local recurrence or metastasis (NEDrl or NEDm); (3) alive with disease, due to presence of local recurrence or metastasis (AWD); (4) dead of disease (DWD). Survival was defined as the time from surgery to last follow-up or death. Complications were recorded and graded according to the Clavien–Dindo classification of surgical complications [8]. In summary, complications were divided in five grades: Grade (I)—Any deviation from the normal postoperative course without the need for pharmacological or interventional treatment; Grade (II)—Requiring pharmacological treatment with drugs; Grade (III)—Requiring surgical, endoscopic or radiological intervention; Grade (IV)—Life-threatening complication requiring intermediate care (IC)/intensive care unit (ICU); Grade (V)—death [8]. Functional results were assessed for all surviving patients using the modified version of the Musculoskeletal Tumor Society (MSTS) functional rating system [9] and Toronto Extremity Salvage Score (TESS) [10].

Statistical analysis. The categorical variables were expressed as percentages of the total patients in a category. The mean, standard deviation and range of all continuous variables were calculated. Survival was analyzed using the Kaplan–Meier analysis. Comparison of the curves was performed in a bivariate analysis with the log-rank test. Differences were considered statistically significant when the p value was less than 0.05. The data were recorded in a Microsoft Excel 2003 spreadsheet and analyzed using Med-Calc Software Version 11.1 (MedCalc Software, Mariakerke, Belgium).

## 3. Results

The study group included 161 patients treated in a single institution: there were 86 males (53%) and 75 females (47%) with a mean age of 56 years (range, 6–76 years) at surgery. Most of these patients were over 18 years of age (144, 89.4%). Almost half of the study population (45%) had at least one comorbidity— in order of frequency: hypertension (39%), cardiovascular disease or congestive heart failure (19%), diabetes (14%), cerebrovascular disease (8%), thyroid disease (7%), chronic obstructive pulmonary disease (5%), vascular disease (4%) and other less representative diseases. The demographic data are summarized in Table 1.

Most patients were treated for their primary tumor (120, 74.5%), whereas 41 patients were referred after unplanned excision or tumor recurrence. Tumors were located in the lower limbs (59–36.6%), upper limbs (31–19.2%), head and neck (34–21.1), trunk (24–14.9) and pelvis (13–8.1%) (Table 2).

Only 38 (23.6%) patients had superficial tumors. The mean tumor volume was 305 mL (median 108 mL, range 3–4082 mL), of which a mean of 175 mL was measured for bone tumors and 625 mL for soft tissue tumors.

Definitive margins were adequate (wide) in 87% of patients with soft tissue tumors. Patients with bone tumors were treated with surgical resection, achieving wide margins in 23/25 patients (92%), a marginal margin in one clavicle resection and a voluntary intralesional margin in one patient with metastatic disease. Bone reconstructions were performed with modular mega-prostheses in all cases with involvement of extremities and periacetabular area, autologous vascularized fibular graft in mandible and bridging osteosynthesis in three patients with thoracic cage involvement. The vast majority of soft tissue reconstructions (96.9%) were performed immediately after tumor resection as part of the same operation, whereas in five cases, they were performed within one week due to long duration of surgery and instability of patient’s conditions. Orthoplastic synergy was considered for soft tissue reconstructions with flaps (31 pedicled flaps and 23 microvascular free flaps) in 54 patients (33.5%), skin grafts in 42 (26.1%), wide surgical excision in 40 patients (24.8%), dermal substitute in 9 (5.6%), incisional biopsy in previous flap reconstructions in 7 cases, allogenic skin grafts in 7 cases and other procedures in the remaining two patients. The overall surgical procedures and flaps performed in the study group are described in Table 3. Vascular bypass was performed simultaneously in 9 patients to restore blood flow to the distal extremity after tumor excision. Adjuvant treatments include radiotherapy in 125 patients (77.6%) and chemotherapy in 27 patients (16.7%).

At a mean follow-up of 5.3 years (range 2–10.5 years), patients with oncologic disease were continuously NED in 130 cases (80.7%), AWD in 10 cases (6.2%) and DWD in 21 cases (13.0%). Eleven patients developed local recurrence and 16 patients developed distant metastases (six of these had both and died of disease). Local recurrence occurred at a mean of 18 months (median 10 months, range 1–78 months), reflecting an aggressive behavior of the neoplasm. Sixty-two patients (38.5%) developed a postoperative complication in this series: 56 minor complications (Clavien–Dindo grades I and II; 90.3%) and 6 major complications (Clavien–Dindo grades III and IV; 9.7%). No patient died of intra/perioperative complications. Total or partial flap loss occurred in 5/48 patients (10.4%). Details of the complications are listed in Table 4.

No differences in terms of complications rate, flap lost and morphological results were noticed between pedicled or free perforator flaps. After surgery, all patients experienced improvement in quality of life resulting from reduction in or resolving of pain and tumor removal. The recovery of the affected limb function was assessed according to the MSTS score in all alive patients at the final follow-up, except for 29 patients with involvement of the abdomen, trunk, thoracic cage and mandible. The mean preoperative MSTS and TESS scores were 62.1 ± 23 and 69.4 ± 13, respectively, which improved up to 74.8 ± 14 and 79.1 ± 13, respectively, at final follow-up.

## 4. Discussion

The “orthoplastic approach” was first coined in the early 1990s by Dr. L. Scott Levin, describing the collaboration between orthopedic and plastic surgeons in limb salvage [11]. Some studies have confirmed that the orthoplastic approach improves outcomes in traumatic patients compared with the historical “piece-meal” approach [12,13]. This new way of collaboration between orthopedic and plastic surgeons is likely to also be effective in other fields such as musculoskeletal oncology. The improvement of surgical techniques and the multidisciplinary approach in musculoskeletal oncology increased the use of complex soft tissue reconstructions and, in some cases, the possibility of a limb salvage approach [14,15,16,17,18,19]. However, as of today, few studies have assessed the efficacy of concerted care between orthopedic and plastic surgeons for complex reconstructions [15]. We therefore sought to study a relatively large number of patients (161 cases) treated in a tertiary center with high expertise in musculoskeletal oncology, finding satisfactory results in terms of oncologic outcomes and also good functional results in demanding limb salvage procedures.

This study has some limitations. (1) We included all patients treated in a multidisciplinary surgical team, analyzing and comparing patients with reconstruction of different sites. Moreover, the heterogeneity of the patient population, as well as different tumor histotypes and the need of adjuvant treatments, should be considered a main limitation. On the other hand, the rarity of primary tumors in specific sites (such as the pelvis or distal extremities) and the long-term experience of a specialized center may increase the value of our study. (2) Although this study is one of the largest series reporting orthoplastic synergy in musculoskeletal oncology, it has only 161 patients with adequate follow-up. Because of the relatively small number of patients in some of our subgroups, we could not analyze all the confounding variables with a multivariate regression model; in fact, we had the choice to reduce the number of variables to increase the value of our analysis. (3) There are no adequate scores to evaluate and compare the quality of life and functional results for such different surgical procedures and sites. However, we decided to use the MSTS score, which is the most used score in musculoskeletal oncology for extremities, associated with a patient-derived measure of physical disability (Toronto Extremity Salvage Score).

The expected benefits of orthoplasty include the possibility of soft tissue reconstructions using the wide surgical techniques in the plastic surgery armamentarium, improvement in quality and healing time of surgical wounds and reduction in postoperative (early and late) complications. On the other hand, the main objective of surgery in musculoskeletal oncology is tumor removal with adequate margins. There is an advantageous theoretical concept in dividing the surgical times of resection and reconstruction, entrusting them to two distinct surgeons: the orthopedic oncologist should focus his/her surgical efforts on obtaining the best resection margins, leaving the concern of soft tissue coverage, aesthetic and functional reconstruction to the plastic surgeon.

Most of the patients with STS require adjuvant radiation therapy (RT) as a standard of care to reduce the local recurrence rates independently to surgical margins [20]. However, both RT and multiple-agent perioperative chemotherapy have led to a non-negligible percentage of local complications (wound dehiscence and infection) [21]. The correct timing of the use of RT compared to surgery is still debated in the literature [22]. Some authors prefer, when feasible, a post-operative RT compared with neoadjuvant radiation to reduce the complication rate [18,19,23,24,25]. In the past, some authors have reported higher wound complication rates in patients who have received radiation, but in many cases, a primary closing of the wounds was performed [25,26,27]. In this view, a tension-free repair, by using a vascularized tissue flap, should be helpful to withstand the radiotherapy, with regular checks to face breakdown of the wound by stopping adjuvant treatment [15,17,19,28]. Others prefer preoperative chemo-radiation and flap reconstruction to avoid delayed adjuvant therapy due to prolonged wound healing when flap reconstruction is performed or a risk of wound breakdown during RT needing secondary surgery [29]. Shaw et al. reported that the waiting time between the surgery and the start of RT should not exceed 31 days [30]. Suh et al. provided RT within 3–4 weeks after soft tissue reconstruction with flaps [19]. In our experience, the mean flap healing time was 23.3 days and the radiation started at a mean of two weeks after flap healing (35.4 days, range 33–46 days).

Another important field of orthoplasty in musculoskeletal oncology is the adequate coverage of prosthetic implants. Modular conventional endoprostheses are commonly used for large bone defects, considering good results in terms of complication rate and restoration of limb function [5,6]. In proximal tibia reconstructions, covering the prosthesis with the medial gastrocnemius muscle flap and reattaching the extensor mechanism on it is currently considered the gold standard [31,32]. Currently, the development of 3D printing technology has provided the possibility to create personalized custom-made prostheses for difficult sites or challenging reconstructions [33,34,35,36]. However, infection remains the main complication and good soft tissue coverage of the prosthesis is considered one of the most relevant factors associated with implant survival [34,37]. This may reflect the increased complexity of cases that required use of modular prostheses.

Is there a correlation between flap reconstructions and oncologic outcomes? Some authors sustain that inherent difficulties in flap reconstruction and possible complications related with surgery may have an added negative impact on the patient outcomes [14]. On the other hand, patients treated with flap reconstruction had better local control than patients treated with primary closure, considering the limits of selection bias and correct surgical indications [15,16,19,38]. Kang et al. reported that patients treated with flap reconstruction had better local control and oncologic outcome than those with primary closure, although they had increased morbidity [15]. This positive prognostic aspect could be justified by the greater possibilities of achieving adequate margins when using an orthoplastic team approach [16,19,38].

Numerous soft tissue reconstruction strategies have been described, from simpler reconstructive options (i.e., split-thickness skin grafts) to complex techniques such as flaps and free tissue transfer [11,39,40,41,42,43]. Split-thickness skin grafts should be avoided, especially after RT, due to the high risk of complications such as chronic draining wound [25,44]. Local flaps (advancement, rotation and transposition) are mainly based on random vascularization in the base of the flap, while local perforator flaps (such as propeller flaps) have a pedicle and can be used by freeing a layer of tissue to fill small defects with exposed vital structures [19,42,45]. When a perforator-based local flap should be used, it is important to avoid a perforator located inside the radiated zone (Figure 1).

Regional flaps provide good solutions for not immediately adjacent small/moderate defects (Figure 2), whereas distal flaps could be used when the donor site is far from the defect, cutting and then re-attaching, micro-surgically, the blood supply at the recipient site (Figure 3).

Soft tissue reconstruction can be performed simultaneously (immediate or within a three-week timeframe) or delayed (after 3 weeks) with respect to surgical tumor resection [17]. Lawrenz et al. specifically analyzed the wound complication rates and oncologic outcomes in a series of 81 patients undergoing immediate (26 cases) versus staged (55 cases) soft tissue reconstruction after soft tissue sarcoma resection [46]. They found similar wound complication rates regardless of timing for reconstruction and advocated staged reconstructions, considering that patients required fewer surgical intensive care unit stays and may maintain the opportunity for reintervention after positive margins with little additional morbidity [46]. Other studies reported advantages of delayed reconstruction in specific settings with uncertain diagnosis or suspicion of intralesional margins [24,47,48]. We do not agree with this staged approach; in fact, in our center, reconstruction is usually planned immediately after resection during the same surgical operation or a few days after, based on the patient’s general health status. Similar results have been reported in a specialized center with a multidisciplinary approach (series of 127 patients with sarcoma resection), where the authors reported 100% wound complications in delayed reconstructions versus 36.7% wound complications in patients treated with immediate reconstruction [49]. In both their and our experiences, the overall incidence of complications was 38% [49]. Few studies in the literature clearly reported the incidence of wound infection without orthoplasty, finding older age, large tumor size, deep tumors, preoperative radiation and previously unplanned surgeries as negative prognostic factors [25,27,49,50,51,52]. An estimated 16–53% [27,49,50,51] of soft tissue sarcoma resections develop complications that require further treatments. A reconstruction after scar formation or after several days of Negative Pressure Wound Therapy (NPWT) poses problems in the isolation of recipient vessels and tissue layers, as well as possible increased risk of infection or in allowing suitable condition for tumor recurrences (Figure 4) [53].

Moreover, the complete histopathologic evaluation of the margin analysis could take several weeks and the risk that such a reconstruction might have to be revised is quite low in an experienced center. Similarly, other studies support the theory that immediate reconstruction may have favorable effects on wound healing, with few complications or need of further surgery [3,17,18,26,27,54].

## 5. Conclusions

Orthoplastic synergy is an effective collaborative combined approach in the therapeutic management of bone and soft tissue sarcomas, maximizing adequate surgical resection, limb salvage and functional recovery. Plastic surgery contributes by offering numerous surgical strategies from skin grafts to microsurgical reconstruction using free tissue transfer and perforators pedicle flaps, with acceptable morbidities. In our experience, immediate reconstructions are technically feasible in one stage, allowing for preservation of limb function and good oncologic outcome, and were not associated with an increased risk of complications. We strongly suggest this close collaboration in specialized high-volume tumor centers as an integrated part of the multidisciplinary care protocol of cancer patients with sarcomas.

## Figures and Tables

**Figure 1 jpm-10-00278-f001:**
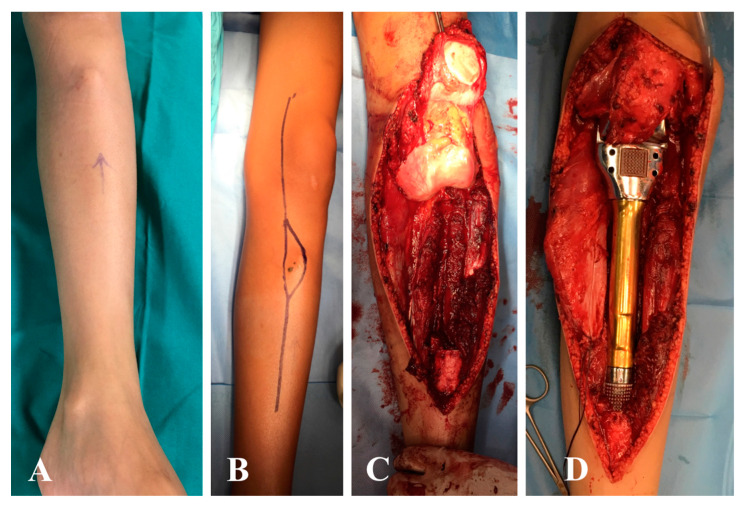
A 12-year-old male with osteosarcoma of the proximal tibia. (**A**) Swelling is visible in the soft tissue near the tibial tuberosity. (**B**) Surgical approach including the needle biopsy tract. (**C**) Surgical field after bone tumor resection and (**D**) after reconstruction with custom-made expandable prosthesis. (**E**) Distal detachment and rotation of the medial gastrocnemius muscle; (**F**,**G**) coverage of the prosthesis with the medial gastrocnemius and soleus muscle, and attachment of the extensor mechanism with sutures to the medial gastrocnemius. (**H**) Wound closure at the end of surgery.

**Figure 2 jpm-10-00278-f002:**
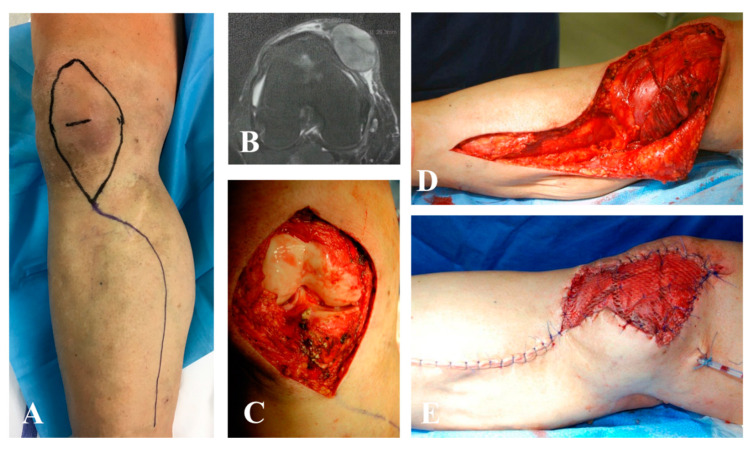
Wide resection for pleomorphic sarcoma of the right knee after radiotherapy and reconstruction with a pedicled medial gastrocnemius muscle flap and skin graft. (**A**,**B**) Preoperative planning is important to avoid basing the flap on perforator vessels located inside the irradiated field. In this case, a muscle flap from the posterior lodge was harvested. (**C**) Surgical field after wide excision of medial ligament and articular capsule; the defect resulted in the exposition of the knee joint, (**D**) requiring a stable and safe reconstruction with a muscle flap. (**E**) The area was covered by partial thickness skin grafts. (**F**) Clinical results at one-year follow-up.

**Figure 3 jpm-10-00278-f003:**
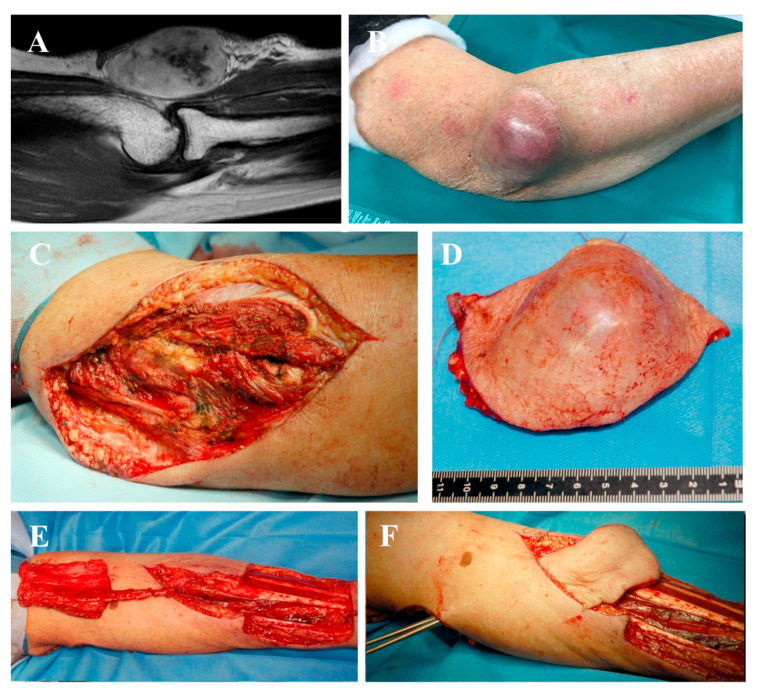
Synovial sarcoma resection after radiotherapy, treated with one-stage reconstruction. (**A**) Preoperative magnetic resonance imaging (MRI) of the affected elbow. When a pedicled or free perforator flap was required, the vascular network was studied with particular attention. (**B**) Clinical aspect of the skin with the soft tissue mass. (**C**) Surgical field after tumor excision showing extensive soft tissue loss, (**D**) corresponding to the specimen. (**E**) The flap was harvested in the forearm and (**F**) passed under a cutaneous bridge to reach the residual defect. (**G**,**H**) Clinical results at three years of follow-up.

**Figure 4 jpm-10-00278-f004:**
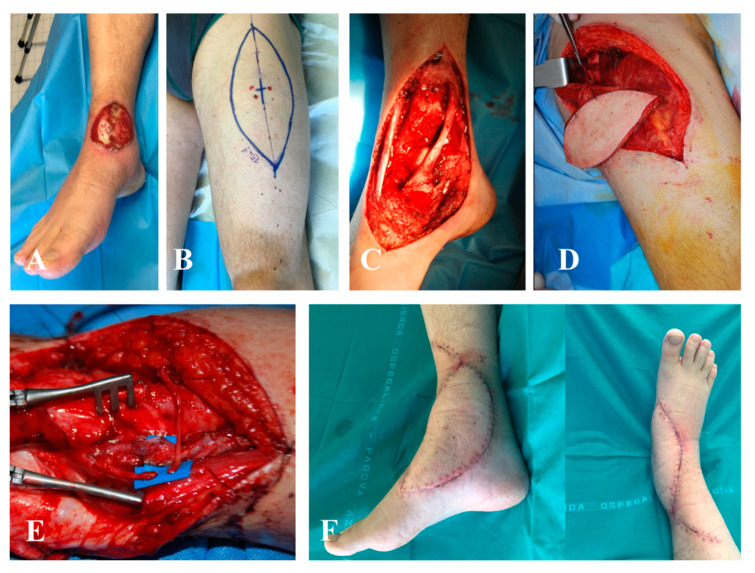
Case of an unplanned excision of sarcoma of the ankle. Wide margin resection of the tumor bed was required. (**A**) Preoperative defect of the ankle and (**B**) the Doppler map of antero-lateral thigh (ALT) perforators. (**C**) Particular of the wide margin excision and (**D**) of the positioning of the ALT free flap to reconstruct the area. (**E**) Microsurgical particular of the arterial anastomosis. (**F**) Follow-up at 3 months.

**Table 1 jpm-10-00278-t001:** Demographic data of entire series (*n* = 161 patients).

Data	Patients	%	Age < 18	Age > 18
Age and gender				
Age (mean years)	56 (range, 6–76)			
Age < 18 years				
Gender (male/female)	86/75			
Pathological diagnosis				
Soft tissue sarcomas	136	84.5%	14	122
Bone tumors	25	15.5%	3	22
Pre-surgical status				
Primarily treated	120	74.5%	16	104
Referral after unplanned excision	24	14.9%	1	23
Referral due to local recurrence	17	10.6%	-	17
Comorbidities				
Hypertension	62	38.5%	-	62
Cardiovascular disease or congestive heart failure	31	19.3%	-	31
Diabetes	23	14.3%	2	21
Cerebrovascular disease	13	8.1%	-	13
Thyroid disease	11	6.8%	1	10
Chronic obstructive pulmonary disease	8	5.0%	-	8
Vascular disease	6	3.7%	-	6
Other less representative diseases	14	8.7%	3	11

**Table 2 jpm-10-00278-t002:** Types and anatomical distribution of soft tissue sarcoma (STS) and bone tumors.

Data	Patients	%	Age < 18	Age > 18
Soft tissue sarcoma (STS)	136/161	84.5%		
- Further treatments in pts with personal history of STS	34/161	21.1%	1	33
- Head, face and neck	28/161	17.4%	2	26
- Lower limb, including hip	21/161	13.0%	6	15
- Upper limb, including shoulder	18/161	11.2%	4	14
- Trunk	6/161	3.7%	-	6
- Thorax	6/161	3.7%	1	5
- Abdomen	5/161	8.1%	-	5
- Pelvis	3/161	1.8%	-	3
- Kaposi’s sarcoma	5/161	8.1%	-	5
- Other body area	10/161	6.2%	-	10
Bone tumors	25/161	15.5%		
- Further treatment in pts with personal history of bone tumors	7/161	4.3%	1	6
- Lower limb	3/161	1.8%	2	1
- Rib, sternum and clavicle	3/161	1.8%	-	3
- Mandible	6/161	3.7%	-	6
- Bone metastasis from carcinoma	6/161	3.7%	-	6

**Table 3 jpm-10-00278-t003:** Number of cases and relative percentages of each surgical procedure. The complete list of specific types of flaps has been exploded.

Data	Patients	%	Age < 18	Age > 18
Flaps	54/161	33.5%		
- Pedicled flaps	31/54	57.4%		
Gastrocnemius	3	5.6%	2	1
Latissimus dorsi	7	13.0%	-	7
Radial forearm	4	7.4%		3
Anterolateral thigh	2	3.7%	1	2
Rectus abdominus	4	7.4%	-	4
Cutaneous perforator propeller	2	3.7%	-	1
Gluteus maximus	2	3.7%	1	2
Soleus	2	3.7%	-	2
Tensor fascia lata	3	5.6%	-	3
Pectoralis	1	1.8%	-	1
Gracilis	1	1.8%	-	1
- Microvascular free flaps	23/54	42.6%	-	
Anterolateral thigh	12	23.5%		9
Latissimus dorsi	7	13.0%	3	7
Radial forearm	2	3.7%	-	2
Gracilis	2	3.7%	-	2
Skin grafts	42/161	26.1%	-	38
Wide surgical excision	40/161	24.8%	4	35
Dermal substitute	9/161	5.6%	5	9
Incisional biopsy in previous flaps	7/161	4.3%	-	7
Allogenic skin grafts	7/161	4.3%	-	6
Synthetic mesh prosthesis	1/161	1.7%	1	1
Amputation	1/161	1.7%	-	1

**Table 4 jpm-10-00278-t004:** Complications of 62 patients (38.5%) classified according to the Clavien–Dindo system.

Data	Patients	%
Grade I	48	77.4%
Including wound dehiscence, delayed wound healing, superficial infection, hematoma		
Grade II	8	12.9%
Including infection needing antibiotic administration, wound dehiscence, partial necrosis, medical aspects (anemia, deep vein thrombosis, urinary tract infection)		
Grade III	5	8.1%
Including deep infection, complete wound dehiscence, hematoma, partial or total flap loss		
Grade IV	1	1.6%
Including myocardial infarction, systemic sepsis		
Grade V	-	-%

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
