# Peer review of "One-Stage Soft Tissue Reconstruction Following Sarcoma Excision: A Personalized Multidisciplinary Approach Called “Orthoplasty”"

_jpm, 2020, doi:10.3390/jpm10040278_

Round 1

Reviewer 1 Report

In these study the authors demonstrated , through  retrospective analysis,   that   the  immediate reconstructions in combination with   multidisciplinary team are technically feasible allowing  limb salvage and functional recovery in patients patients with bone and soft tissue sarcomas.

This seems a promising study and the methodology is sufficiently robust.

Author Response

Thank you reviewer 1 for your positive comment.

As suggested, the manuscript has been reviewed by a native English speaker and corrections have been highlighted in red.

Reviewer 2 Report

The authors describe outcomes for patients with sarcoma requiring surgical local control. 

1. Would the authors be able to stratify demographics and outcomes for <18 yrs and >18 years? Chemotherapy and management protocols vary significantly between the adult and pediatric population and may be able to help identify additional supportive factors to surgical approaches taken. 

2. In Table 2, metastasis listed under bone tumors. I am assuming this is extrapulmonary metastasis, please specify as such.

3. Results, line 183: Can the authors elaborate of time from surgery to local recurrence? Earlier recurrences may be indicative of poor surgical control with narrow margins, or need for post op radiotherapy etc.

4. Is there any data on wound infections or surgical failure rates when performed without an orthoplastic approach? 

Author Response

Response to Reviewer 2 Comments

Point 1:The authors describe outcomes for patients with sarcoma requiring surgical local control. Would the authors be able to stratify demographics and outcomes for <18 yrs and >18 years? Chemotherapy and management protocols vary significantly between the adult and pediatric population and may be able to help identify additional supportive factors to surgical approaches taken.

Response 1: Thank you reviewer 2 for your interesting comment. We agree with the need of patient stratification considering the age. Considering the majority of soft tissue neoplasms in the present series, obviously most of the patients were over 18 years of age (144, 89.4%). Therefore, it is not possible to perform a statistical analysis on the outcomes. Looking the literature, chemotherapy seems not to be a statistically significant factor with regard to wound complications after sarcoma resection (Arbeit JM et al. J Clin Oncol. 1987; Devereux D et al. Cancer 1980; Sanniec KJ et al. Ann Plast Surg. 2013).

However, we updated the tables including the patient’ stratification based on “18 years-old” as cut-off. With the limited power analysis, no differences were found on complications and outcomes.

Point 2:In Table 2, metastasis listed under bone tumors. I am assuming this is extrapulmonary metastasis, please specify as such.

Response 2:Thank you reviewer 2. we changed the non-specific term of "metastasis" to the more correct one of "bone metastasis from carcinoma"

Point 3:Results, line 183: Can the authors elaborate of time from surgery to local recurrence? Earlier recurrences may be indicative of poor surgical control with narrow margins, or need for post op radiotherapy etc.

Response 3:Thank you reviewer 2. Eleven patients developed a local recurrence at a mean of 18 months (median 10 months, range 1 month-78 months). Distant metastases occurred also in six of them, reflecting an aggressive behaviour of the neoplasm. As reported in pag 7 line 160 “Definitive margins were adequate (wide) in 87% of patients with soft tissue tumors. Patients with bone tumours were treated with surgical resection achieving wide margins in 23/25 patients (92%), marginal margin in one clavicle resection and voluntary intralesional margin in one patient with metastatic disease.”.

Point 4:Is there any data on wound infections or surgical failure rates when performed without an orthoplastic approach?

Response 4:Thank you reviewer 2. Few studies in literature clearly reported the incidence of wound infection without orthoplasty, finding older age, large tumor size, deep tumors, preoperative radiation and previously unplanned surgeries as negative prognostic factors. An estimated 16-53% of soft tissue sarcoma resections develop complications that require further treatments. In our experience and in similar papers the wound complication rates was about 38%.

We added these data in the text and updated the reference list with related citations:

  1. Sanniec KJ, et al. Predictive factors of wound complications after sarcoma resection requiring plastic surgeon involvement. doi: 10.1097/SAP.0b013e31827c7973.
  2. Nakamura T, et a. Clinical characteristics of patients with large and deep soft tissue sarcomas. doi: 10.3892/ol.2015.3289.
  3. Saddegh MK & Bauer HC. Wound complication in surgery of soft tissue sarcoma. Analysis of 103 consecutive patients managed without adjuvant therapy. PMID: 8472424.
  4. Geller DS, et a.. Soft tissue sarcoma resection volume associated with wound-healing complications. doi:10.1097/blo.0b013e3180514c50.